# Developing a risk prediction model for death at first suicide attempt—Identifying risk factors from Thailand's national suicide surveillance system data

Suwanna Arunpongpaisal[1,2☯], Sawitri Assanangkornchai[1☯]*, Virasakdi Chongsuvivatwong[1☯]

1 Department of Epidemiology, Faculty of Medicine, Prince of Songkla University, Hat Yai, Songkhla, Thailand, 2 Faculty of Medicine, Khon Kaen University, Khon Kaen, Thailand

☯ These authors contributed equally to this work.
* savitree.a@psu.ac.th

## Abstract

More than 60% of suicides globally are estimated to take place in low- and middle-income nations. Prior research on suicide has indicated that over 50% of those who die by suicide do so on their first attempt. Nevertheless, there is a dearth of knowledge on the attributes of individuals who die on their first attempt and the factors that can predict mortality on the first attempt in these regions. The objective of this study was to create an individual-level risk-prediction model for mortality on the first suicide attempt. We analyzed records of individuals' first suicide attempts that occurred between May 1, 2017, and April 30, 2018, from the national suicide surveillance system, which includes all of the provinces of Thailand. Subsequently, a risk-prediction model for mortality on the first suicide attempt was constructed utilizing multivariable logistic regression and presented through a web-based application. The model's performance was assessed by calculating the area under the receiver operating curve (AUC), as well as measuring its sensitivity, specificity, and accuracy. Out of the 3,324 individuals who made their first suicide attempt, 50.5% of them died as a result of that effort. Nine out of the 21 potential predictors demonstrated the greatest predictive capability. These included male sex, age over 50 years old, unemployment, having a depressive disorder, having a psychotic illness, experiencing interpersonal problems such as being aggressively criticized or desiring plentiful attention, having suicidal intent, and displaying suicidal warning signals. The model demonstrated a good predictive capability, with an AUC of 0.902, a sensitivity of 84.65%, a specificity of 82.66%, and an accuracy of 83.63%. The implementation of this predictive model can assist physicians in conducting comprehensive evaluations of suicide risk in clinical settings and devising treatment plans for preventive intervention.

**Data Availability Statement:** All relevant data are within the paper and its Supporting information files.

**Funding:** The author(s) received no specific funding for this work.

**Competing interests:** The authors have declared that no competing interests exist.

## Introduction

Suicide and suicide attempts are major public health concerns, causing a great deal of emotional and economic burden not only to families and close relatives but also to the community [1, 2]. Most previous studies focusing on risk factors for suicide have found a previous suicide attempt to be a strong risk factor for suicide mortality [3]. However, more than half of suicide victims die on their first attempt [4, 5]. This implies that a sizable fraction of the population that commits suicide goes unrecognized in studies examining this problem.

### Risk factors for suicide

The causation of suicide is related to genetics, demographics, culture, geography, and personal experience [6]. Various factors can increase an individual's risk of suicide, such as being older, male, unemployed [7], having a previous suicide attempt [8], multiple childhood adversities and/or trauma, such as parental neglect, physical and/or sexual abuse [9, 10], financial difficulties [11], exposure to family violence [12], or intimate partner relationship problems [13], having a physical illness resulting in severe disability and/or chronic pain, such as traumatic brain injury victims [14], mental disorders, particularly mood disorders [15], schizophrenia [16], and alcohol/substance use disorders [17, 18]. Some prior research studies have identified several factors that are linked to a lower risk of suicide for individuals. These factors include resilience to stress, optimism, extroversion, religiosity, a strong sense of purpose and self-worth, responsibility for young children, and efficient palliative care for the elderly [6].

A recent umbrella review identified a wide range of individual-level risk factors for suicide mortality in the general population, most of which were from high-income countries. A previous suicide attempt, suicidal ideation and psychiatric disorders were found to be associated with a greatly increased risk of suicide mortality. Physical illnesses, such as cancer and epilepsy and sociodemographic factors (including unemployment and low education), contact with the criminal justice system, state care in childhood, access to firearms, and parental death by suicide also increased the risk of suicide mortality [3].

In addition, numerous risk factors for suicide might be related to one another [6], and the consequences of these factors may vary depending on the environment in which they occur. For example, losing one's work or experiencing financial difficulties can have an impact on an individual and subsequently cause their relationships to worsen. On a systemic level, these consequences can also be connected to an economic downturn. Therefore, in both research and clinical practice, identifying combinations of factors that can act in concert to increase the risk of suicide may prove to be more beneficial than listing individual ones.

Despite abundant research on risk factors for suicide, there is a dearth of research on the factors that can predict the initial occurrence of suicidal thoughts, plans, attempts, and deaths as well as the reliability of these predictors, especially in low- and middle-income countries. A few studies on first-onset suicidal thoughts and behaviors have been conducted among college students as this group appears to be at the greatest risk of developing first-onset suicidal thoughts and behaviors. Two studies among college students found that adverse experiences such as dating violence and physical abuse before the age of 17, betrayal by someone other than the partner [19], ongoing arguments or breakup with a romantic partner, emotional abuse, parental death and lack of a reliable support system [20] were strong predictors for first-onset suicidal thoughts and behaviors.

Moreover, a secondary data analysis among individuals aged 18–89 years old from the National Violent Death Reporting System spanning from 2005 to 2013 in the USA found that 79% of these individuals died by suicide on their first suicide attempt. Compared to those with a history of suicide attempts, those who died on their first attempt were more likely to be of the

male gender, married, of African-American ethnicity, and above the age of 64. Those first-attempt decedents were also more inclined to employ extremely lethal means, less likely to have a documented mental health condition or to have communicated their intentions to others, and more likely to die if there is a physical health issue or a criminal or legal issue [5].

## Prediction models for suicide

A prediction model is a mathematical process used to predict future events or outcomes by analyzing patterns in a given set of risk factors [21]. A significant percentage of suicide victims interact with the medical system in the year before their death [22, 23], thus several suicide prediction systems have been developed as a result of this fact to help identify patients who should receive preventive measures. The main aim of a suicide prediction model is to enhance our capacity to predict and therefore prevent suicide, beyond the existing clinical level, by supporting clinicians in their decision-making. However, low sensitivities and low positive predictive values have led critics to argue that these tools have little clinical value and, in some cases, might do more harm than good [24, 25]. A systematic review of unique prediction models across five countries, the United States of America, Australia, the United Kingdom, Iran, and South Korea, found that the global classification accuracy of most suicide prediction models was good ($\geq 0.80$); however, their accuracy in predicting future events was extremely low ($\leq 0.01$ in most models), limiting their usability in clinical settings [26].

Some earlier studies have started with a predictor set made up of sociodemographic and clinical data taken from medical records, occasionally supplemented with a set of clinician rating scales and patient self-reports. Multivariate analyses using these data and various types of machine-learning methods have been used to develop prediction tools in more recent studies [27, 28]. However, because of the low base rate of suicides, it is hard to create a sufficiently large data set for analysis to test complex models or for the estimation of a trustworthy machine-learning algorithm.

Most related studies in this field have developed suicide prediction tools for three high-risk groups: people who are suicidal and have been taken to an emergency room; psychiatric inpatients while they are in the hospital; and people who have been discharged from the hospital following a suicide attempt [29–31]. Little research has been conducted on prediction tools for the first suicide attempt and its outcomes.

## Suicide surveillance system in Thailand

Reported suicide rates in Thailand are lower than the global average, according to the World Health Organization suicide database. In 2017, Thailand reported a suicide rate of 6.03/100,000, while the global average was 6.5/100,000 [32, 33]. One likely reason for the lower rate in Thailand is that the estimations of the suicide rate at the national level were based solely on the national death certification system, of which the proportion of ill-defined conditions was nearly 50%, as many deaths in Thailand occurred at home, and the causes of death were reported by medically untrained local officers [34].

In 1999, a Burden of Disease and Risk Factors study from Thailand reported that suicide was one of the top 20 leading risk factors for disability-adjusted life years (DALYs) in men, accounting for 2.7% of all DALYs and 4.9% of deaths in men [35]. The Department of Mental Health of the Ministry of Public Health mandated that the Khon Kaen Rajanagarindra Psychiatric Hospital develop a National Suicide Prevention Center and suicide surveillance system in 2001. Under this system, a self-directed violence-506S (SDV-506S) case registration system with standard operating procedures was established to register all suicide attempt cases in 33 pilot community hospitals across four regions of the country, which was then expanded to all

76 provinces in 2005. The SDV 506S registration form was revised, and an online registration system was developed in 2017.

More than 60% of suicides worldwide are believed to occur in low- and middle-income countries (LMIC) [36]. Regrettably, there is less knowledge regarding the characteristics of individuals who die by suicide on their first attempt, as well as the factors that increase the likelihood of first-attempt suicide mortality in these areas [5, 37, 38]. Identifying factors related to suicide among this understudied group could help efforts to identify more people who may be at risk for suicide and develop preventive measures.

This study extends the previous literature by describing the characteristics of first-suicide attempters and identifying predictors of mortality in first-attempt suicide in a low-middle-income country, using data from the National Suicide Surveillance System. To assist clinicians in assessing suicide risk in their clinical practice, we also created a risk prediction model and smartphone application and tested their validity based on the identified risk factors.

## Materials and methods

This study was an analysis of data from the National Suicide Surveillance System based on all the SDV-506S records of Thai individuals who first attempted suicide between May 1, 2017, and April 30, 2018.

Each record in the SDV-506S database comprises sociodemographic and clinical characteristics of the decedents, information about suicidal behaviors, and precipitating factors. This information is obtained through medical interviews and physical and mental health examinations of the patients and their relative(s) when the patient visits the emergency department or is admitted to the hospital. The classifications of both suicide and intentional self-harm follow the International Classification of Disease, Version 10 (ICD-10) codes X60-X84, Z91.5, and R45.8, to ensure the uniformity and consistency of recorded information internationally.

All cases of suicide attempts in public healthcare settings and suicide decedents who die at the hospital are recorded by the hospital staff after immediate care. For those who die unnaturally at home, police authorities are legally empowered to request that government medical institutions in the area perform medical forensic autopsies of suspected unnatural death cases. In cases of suspected suicide, a verbal autopsy is conducted by health professionals in the area, and their findings are recorded in the system. The revised national suicide surveillance system provides the best possible data source for studying suicide risk factors because of its large size and national representation.

No identification data for the patients were included in the dataset provided to us. After excluding 74 duplicate records, 3,324 records were included in this study's analysis. Data analysis for this study was carried out from March to April 2023.

### Variables examined

Overall, 22 variables were included in the analysis (Table 1). The main outcome variable was the suicidal outcome, namely, death or survival. The potential predictive variables for mortality at the first suicide attempt were grouped into five domains: 1) demographics (four variables), 2) underlying disease(s) (six variables), 3) interpersonal problems (seven variables), 4) financial problems (two variables), and 5) suicidal attributes (two variables). We did not consider the lethality of the suicidal actions in the model, as lethality is considered a variable of the method of suicidal action rather than a predictive variable of suicide. Psychiatric and physical disorders were diagnosed by medical doctors in a healthcare setting where patients were sent for treatment of health conditions resulting from their suicide attempts. Alcohol- and substance-related problems were identified based on the results of screening by healthcare

**Table 1. Variables included in the analysis.**

| Domain | Variables and definitions |
|---|---|
| **1. Demographics** | 1. Sex: (male, female); 2. Age group: (10–24, 25–50, >50) 3. Marital status: (has a spouse, no spouse); 4. Working status: (working, not working). |
| **2. Underlying diseases** | Psychiatric disorders: 5. Depressive disorder(s): (yes/no); 6. Psychotic disorder(s): (yes/no); 7. Other psychiatric disorder(s) (having at least one other psychiatric disorder or condition, for example, anxiety disorder, stress-related disorder, insomnia, bipolar disorder, gambling disorder, or mental retardation): (yes/no); 8. Alcohol-related problem: (yes/no); 9. Substance-related problem: (yes/no); 10. Chronic physical illness: (having at least one chronic physical illness, for example, a noncommunicable disease (NCD), cancer, HIV, etc.: (yes/no). |
| **3. Interpersonal problems** | 11. Being strongly criticized, blamed, or intimidated: (yes/no); 12. Lovesickness or jealousy: (yes/no); 13. Desire for plentiful attention: (yes/no); 14. Quarrel with significant person(s): (yes/no); 15. Conflicts with work colleague(s): (yes/no); 16. Being neglected: (yes/no); 17. Recent loss of a significant person: (yes, no). |
| **4. Financial problems** | 17. Losing in business: (yes/no); 18. Being in big debt: (yes/no). |
| **5. Suicidal attributes** | 19. Method of suicide attempt: a. high-lethality method, such as by gun, hanging, drowning, jumping from a height, gas inhalation, flames, or car crash; b. low lethality methods, such as drug overdose, chemical liquid poisoning, pesticide, herbicide, alcohol/organic solvent toxicity, cutting/blunt injury, and other/unspecified means, 20. Having suicidal intention: (yes/no); 21. Having warning signs of suicide: (yes/no). |
| **Suicide outcome** | 22. Death or survival |

providers using the Alcohol Use Disorder Identification Test and the V2-Thai Substance Screening Test [39]. All variables except age group were coded as binary (yes/no).

## Statistical analysis

First, we described the characteristics of all first suicide attempters (N = 3324) in terms of their demographics and suicidal methods. Second, we compared all variables by suicide outcome (death or survival) of these attempters. We then split the dataset into two randomly-selected subsets, the first for developing a predictive model which included 1,824 records and the second for evaluating the model's predictive capability with the remaining 1,500 records.

For the model development, initially, bivariate associations between each independent variable and the suicidal outcomes were analyzed using the Chi-square test or Fisher's exact test, as appropriate. We then performed multivariable logistic regression modelling separately for three predictive domains: 1) demographics, 2) underlying diseases, and 3) interpersonal and financial problems and suicide attributes. Variables with p-values less than 0.05 from all domains were then included in the combined multivariable logistic regression using the first subset of 1,824 records. All significant independent variables ($p < 0.05$) were included in the final risk prediction model.

Next, to perform model validation, the prediction model derived from the first dataset was run on the second dataset of 1,500 records. The probabilities derived from the risk prediction model for all individuals were compared with those estimated using a logistic model. We then used a receiver operating characteristic (ROC) curve to assess the discriminative power of the risk prediction model, and we calculated the area under the curve (AUC) with a 95% confidence interval (CI). Other model performance metrics, including accuracy, sensitivity and specificity, were also analyzed.

Finally, a risk prediction model relating the risk of an individual experiencing an event (death) to a set of predictors based on the first dataset was calculated using multivariable logistic regression as follows:

Individual risk of death = exponential × (individual risk score) ÷ (1+exponential × (individual risk score), where an individual risk score = (intercept + b1×1 + b2×1 + ...bpxp). b1,

b2, and bp indicate regression coefficients that describe how an individual's values of predictor variables affect the risk of death [40].

The R program (version 4.3.2) was used for all analyses. To make the prediction model available to practicing healthcare personnel, we created a web-based application for calculating the probability of death on a first suicide attempt (https://app.calconic.com/).

The study protocol for this analysis of the SDV-506S data was approved by the Ethics Committee for Research in Human Subjects of the Faculty of Medicine, Prince of Songkla University (REC.62-077-18-1). The requirement for informed consent was waived due to the retrospective nature of the study.

## Results

### Sample characteristics

Of the 3,324 first-time suicide attempters, 2,037 (61.3%) were of the male sex. The mean age was 40.02 years (SD = 17.16); 718 (21.6%) were 10–24 years old, and 895 (26.9%) were ≥ 50 years old. Approximately half (50.8%) were married and living with their spouse; 35.1% were single; the remainder were either widowed, divorced, or separated. Almost all patients (96.7%) were Buddhist. Among all cases, 70.5% were employed, 17.3% were studying, and 12.2% were unemployed. Approximately half of the attempters used low-lethality methods (53.2%), including drug overdose, poisoning by pesticides, herbicides, or chemical agents, cutting or stabbing, or other means (21.5%, 25.1%, and 5.5% of all attempters, respectively). The most popular high-lethality method among those who used this type of method was hanging (37.4%), then jumping from a height (5.9%), using guns (3.1%), and other methods (0.6%), such as drowning, car crash, or gas inhalation.

### Bivariate and multivariable analyses

The total number of individuals who died on their first attempt was 1,680, accounting for 50.5% of all cases. Table 2 shows the associations between each independent variable and suicide outcomes. Males were more likely to die on their first attempt (65.1% vs. 27.4% for females). Older individuals were also more likely to die than younger ones. Depressive or psychotic disorders and alcohol- and substance-related problems were positively associated with death on the first suicide attempt.

In multivariable logistic regression modelling using the first dataset of 1,824 records, 9 of the 18 potential variables from the bivariate analyses remained significantly associated with death on the first suicide attempt in the final model. These were three demographic variables (sex, age group, and working status), two underlying diseases (depressive disorders and psychotic disorders), two interpersonal problems (being strongly criticized and desire for attention), and two suicidal attributes (suicidal intention and suicidal warning signs).

The estimated regression coefficients for the predictive model based on 1,824 records are presented in Table 3. When this prediction model was tested in the second dataset, the area under the ROC curve based on the second dataset (1,500 records) for the final nine-variable prediction model was 0.902 (95% CI: 0.886, 0.917), and the optimum cut-off point was 52.16 with a sensitivity of 84.65%, a specificity of 82.66%, and an accuracy of 83.63%, indicating a high discriminative ability (Fig 1 and Table 4).

### Risk prediction model for mortality at the first suicide attempt

We calculated the risk scores for each individual and their predicted risk of mortality at the first suicide attempt using the regression coefficients derived from the development dataset as

**Table 2. Socio-demographic, clinical and suicide attributes by suicide outcome.**

| Variable | Total (n) (N = 3,324) | Decedents (%) (N = 1,680) | Survivors (%) (N = 1,644) | P-value |
|---|---|---|---|---|
| **Sex: Female** | 1,286 | 27.4 | 72.6 | <0.001 |
| Male | 2,038 | 65.1 | 34.9 | |
| **Age (years): 10–24** | 717 | 23.3 | 76.7 | <0.001 |
| 25–50 | 1,712 | 50.5 | 49.5 | |
| >50 | 895 | 72.4 | 27.6 | |
| **Marital status:** | | | | 0.168 |
| Having a spouse | 1,689 | 51.7 | 48.3 | |
| No spouse | 1,635 | 49.3 | 50.7 | |
| **Working status: Working** | 2,919 | 44.2 | 55.8 | <0.001 |
| Not working | 405 | 96.0 | 4.0 | |
| **Depressive disorder: No** | 3,175 | 49.2 | 50.8 | <0.001 |
| Yes | 149 | 79.2 | 20.8 | |
| **Psychotic disorder: No** | 2,949 | 49.6 | 50.4 | 0.003 |
| Yes | 375 | 57.9 | 42.1 | |
| **Other psychiatric disorder:** | | | | 0.017 |
| No | 3,027 | 51.0 | 49.0 | |
| Yes | 117 | 39.3 | 60.7 | |
| **Alcohol-related problem: No** | 2,544 | 47.4 | 52.6 | <0.001 |
| Yes | 780 | 60.9 | 39.1 | |
| **Substance-related problem:** | | | | <0.001 |
| No | 3,050 | 49.0 | 51.0 | |
| Yes | 274 | 67.9 | 32.1 | |
| **Physical illness: No** | 2,623 | 44.5 | 55.5 | <0.001 |
| Yes | 701 | 73.0 | 27.0 | |
| **Being criticized: No** | 2,238 | 54.6 | 45.4 | <0.001 |
| Yes | 1,086 | 42.3 | 57.7 | |
| **Lovesickness: No** | 2,780 | 52.6 | 47.4 | <0.001 |
| Yes | 544 | 39.9 | 60.1 | |
| **Desire for attention: No** | 1,321 | 51.7 | 48.3 | < 0.001 |
| Yes | 203 | 32.5 | 67.5 | |
| **Quarrel with significant person: No** | 2,019 | 58.8 | 41.2 | < 0.001 |
| Yes | 1,305 | 37.8 | 62.2 | |
| **Conflict with colleague: No** | 3,268 | 50.9 | 49.1 | 0.001 |
| Yes | 56 | 28.6 | 71.4 | |
| **Being neglected: No** | 3,244 | 50.1 | 49.9 | < 0.001 |
| Yes | 80 | 70.0 | 30.0 | |
| **Recent loss of loved one: No** | 3,247 | 50.4 | 49.6 | 0.198 |
| Yes | 77 | 58.4 | 41.6 | |
| **Losing in business: No** | 2,760 | 48.3 | 51.7 | < 0.001 |
| Yes | 564 | 61.5 | 38.5 | |
| **Being in debt: No** | 3,273 | 50.4 | 49.6 | 0.106 |
| Yes | 51 | 62.7 | 37.3 | |
| **Suicidal intention: No** | 1,394 | 14.4 | 85.6 | <0.001 |
| Yes | 1,930 | 76.6 | 23.4 | |
| **Having warning signs of suicide: No** | 2,515 | 43.2 | 56.8 | <0.001 |
| Yes | 809 | 73.4 | 26.6 | |

**Table 3. Risk prediction model of death at first suicide attempt using multivariable logistic regression (Model development dataset, N = 1,824).**

| Variable | Coefficient | Adjusted OR | 95% CI | P value |
|---|---|---|---|---|
| Intercept | -4.0768 | | | |
| Sex: Male/Female | 1.6854 | 5.39 | 3.99, 7.34 | <0.001 |
| Age (in years): | | | | |
| 25–50/10–24 | 0.8575 | 2.36 | 1.59, 3.51 | <0.001 |
| >50/10–24 | 1.5089 | 4.52 | 2.9, 7.11 | <0.001 |
| Work status: Not working/ working | 3.9269 | 50.75 | 22.35,138.62 | <0.001 |
| Depressive disorder: Yes/No | 0.8711 | 2.39 | 1.25, 4.7 | 0.01 |
| Psychotic disorder: Yes/No | -0.4348 | 0.65 | 0.43, 0.98 | 0.041 |
| Being criticized: Yes/No | -0.4260 | 0.65 | 0.48, 0.88 | 0.006 |
| Desire for attention: Yes/No | -0.7210 | 0.49 | 0.26, 0.89 | 0.019 |
| Suicidal intention: Yes/No | 2.9692 | 19.48 | 14.35, 26.77 | <0.001 |
| Suicide warning signs: Yes/No | 1.0558 | 2.87 | 2.06, 4.05 | <0.001 |

OR: odds ratio, CI: confidence interval

shown in Table 3. To illustrate, we present an example of an individual with a set of predictive variables. A risk score of a male, aged 55, not working, with a depressive disorder, no psychotic disorders, being criticized, no desire for attention, showing some suicidal intention and warning signs could be calculated as: -4.0768 + 1.6854*1(male sex) + 1.5089*1(age-group >50) + 3.9269*1(not working) + 0.8711*1(having depressive disorder)—0.4348*0(no psychotic symptoms)—0.426*1(being criticized)—0.721*0*(no desire for attention) +2.9692*1(having suicidal intention) +1.0558*1(showing warning signs) = 7.5145. Therefore, the predicted risk of mortality at the first suicide attempt was [exp(7.5145) ÷ (1+exp(7.5145))] × 100 = 99.95%.

This web-based risk prediction model can be found at: https://preview.calconic.com/6446184244c67700292bdfd3.

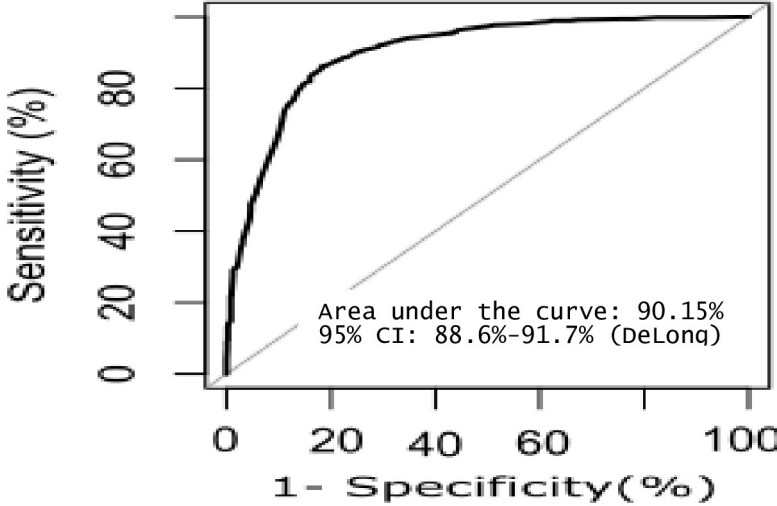

**Fig 1. Receiver operating characteristic (ROC) curve of the risk-prediction model of mortality at the first suicide attempt (Model evaluation dataset, N = 1500).**

**Table 4. Performance metrics of the 9-factor predictive model (Model evaluation dataset, N = 1,500).**

| Death risk score | Sensitivity % | Specificity % | Accuracy % |
|:---:|:---:|:---:|:---:|
| **45.09** | 86.35 | 81.03 | 83.69 |
| **47.27** | 86.22 | 81.30 | 83.75 |
| **49.66** | 85.43 | 81.84 | 83.62 |
| **52.16** | **84.65** | **82.66** | **83.63** |
| **54.03** | 83.07 | 84.01 | 83.50 |
| **56.11** | 82.81 | 84.01 | 83.37 |
| **58.14** | 81.89 | 84.01 | 83.50 |

## Discussion

Our analysis found that in our study cohort, 50.5% of first suicide attempts resulted in death. This proportion is remarkably high, although it falls within the range found in prior studies; around 50–80% of those who die by suicide did so on their first attempt [5, 38]. We also identified nine significant predictive factors for death on the first suicide attempt. To our knowledge, this is the first nationally representative study in an LMIC to report the predictors of death among those with a first suicide attempt. Several of these factors, in particular, being male, being older, having a psychiatric disease, and having an interpersonal problem, are in line with previous studies [7, 13, 37, 41–44]. The web-based risk-scoring tool developed in this study for predicting death on the first suicide attempt is simple, user-friendly, and feasible for use by clinicians in busy clinical settings.

The strongest predictor found in our study was employment status, particularly having no job in the recent period before the suicide attempt. In a rapidly transitioning society such as Thailand, where people are achievement-oriented and compete for success, individuals who have no job or have recently lost a job could be regarded as failures and become socially alienated. Having recently lost a job also engenders other problems such as financial problems, marital problems, loss of self-esteem, and criticism, etc. These people are a vulnerable group in society and are prone to anomie. This vulnerability has the potential to heighten the likelihood of engaging in suicidal behaviors or suffering from other psychiatric diseases [45, 46]. This suggests that preventive interventions targeting unemployed persons, especially those having recently lost their jobs through mental health and suicidal risk screening programs, and providing biopsychosocial interventions with social welfare support to help them reengage in work, may be advantageous to their well-being and social integration, which may reduce any suicidal inclinations.

Our study revealed that having suicidal intentions was a highly reliable indicator of death on the first suicide attempt. Multiple studies have specifically examined suicidal intentions [47–49]. One study reported a positive correlation between age and the intensity of suicidal intent, as well as the presence of warning indicators [50]. This finding is a significant indicator of suicide risk in older persons who might be at risk of attempting suicide. In our study, almost 58% of individuals who attempted suicide for the first time reported having suicidal intentions. With its strong influence on the risk of death (adjusted OR = 19.48) and high prevalence, an in-depth assessment of the degree of suicidal intention and warning signs among those prone to suicidal behaviors, especially older patients, and appropriate interventions could prevent a high proportion of at-risk people from death.

Our findings indicate that two specific interpersonal difficulties, namely experiencing severe criticism and desire for plentiful attention, were negatively associated with the fatality outcome among the first suicide attempters. Previous studies found relationships between

some interpersonal factors, for example, perceived criticism [51], high level of expressed emotion [52], bullying [53], and cyber-victimization [54] with suicide ideation and attempts. These links are often explained by the interpersonal theory of suicide which emphasizes the mediation role of thwarted belongingness and perceived burdensomeness [55, 56]. These relationships are also in line with our study, in which we also found that the experience of being strongly criticized, blamed, or intimidated was common among our subjects (32.67% of all first suicide attempters) and more frequent in the surviving attempters (57.7%) than in the decedents (42.3%). Similarly, a desire for plentiful attention was higher among the surviving attempters (67.5%) than the decedents (32.5%), making it a risk factor for non-fatal suicidal behaviors. Suicide attempts may be an expression of feelings, like emotional pain, sadness, abandonment, or anger, in response to criticism or a desire for plentiful attention from a significant person; thus, their role in the risk prediction model can be a negative risk factor for death. If the significant persons recognize and respond appropriately, suicide attempters may change their suicidal thoughts and want to continue living. Furthermore, cognitive reappraisal or the development of healthy coping skills to deal with criticism and a desire for plentiful attention may be especially important for these people.

Our risk prediction model indicates that being male and older increases an individual's susceptibility to a 29.27% risk of mortality at the first suicide attempt. If the individual possesses additional predisposing characteristics such as unemployment and depression his likelihood of mortality may escalate to the range of 95.45% to 98.05%. Moreover, if an individual possesses suicidal intent and exhibits warning symptoms of suicide, their likelihood of mortality could approach 99.96%. With its high discriminative ability, as shown by the high area under the ROC curve and the high internal validity of the model, our suicide risk-prediction model may have an advantage over traditional tools in Thailand, like the 8 Questions (8Q) for the Suicidality Test [57]. Moreover, it is more convenient and uncomplicated for an emergency physician to evaluate the first suicide attempt and choose appropriate measures to prevent subsequent suicide. However, the model requires external validation to calibrate and confirm its discriminative ability.

The main strength of this study was its large sample size, obtained from a nationwide suicide surveillance system, enabling a high level of accuracy in the findings. Nevertheless, our study does possess certain limitations. The Thai national suicide surveillance system mostly relies on web-based technologies; however, certain data are still transmitted using traditional paper-based techniques. Consequently, the process of reporting data is slow, leading to the possibility of missing and incomplete information. Furthermore, a significant proportion of fatalities in Thailand occur within residential settings, and local officials lacking medical expertise are responsible for completing the corresponding death certificates. A previous study discovered that the rates of accurate reporting for the causes of death among individuals who died in hospital settings and those who died outside of hospitals were 53.4% and 29.7%, respectively [34]. Furthermore, it is possible that certain variables, such as alcohol intake or substance-related disorders, may have been inaccurately recorded or not fully accounted for. Finally, clinical information that is highly relevant to suicide, such as the severity of psychiatric symptoms, was unavailable.

With the high availability of electronic health records (EHR) in modern medical practice, studies of machine-learning methods have grown rapidly as suicide prediction methods [58]. Although the use of machine learning in suicide research is promising for improving the prediction and prevention of suicidal thoughts and behaviors, these machine learning methods require voluminous data to first train the model and a different dataset to evaluate the model. Our national suicide surveillance system has only developed an electronic online system in 2017 and records of the first suicide attempters are still small. Besides, our aim for this research

was not only to develop a prediction model but also to understand and interpret the relationship between predictors and the outcome variable. Therefore, we used logistic regression to develop the prediction algorithm. In the future when our surveillance system is well established and contains a large amount of data, a machine-learning prediction model can be applied for further improving the prediction tool.

This predictive model for mortality in individuals making their first suicide attempt can be extensively utilized in emergency departments and outpatient clinics, including counselling services and inpatient facilities. To enhance the efficacy of suicide prevention initiatives, it is advisable to facilitate greater engagement and intervention for those who have made their first suicide attempt, particularly those with a higher likelihood of actually committing suicide, using the mobile application of this prediction model. To ensure the accuracy and reliability of the predictor model for mortality in individuals making their first suicide attempt, further investigation is required utilizing larger datasets, advanced machine-learning techniques, and rigorous validation methodologies. Moreover, approaches tailored for individual subjects should be implemented in future studies.

## Conclusion

This study involved the development and validation of a risk prediction model for mortality in individuals who had made their first suicide attempt, using nine variables, sex, age group, work status, depressive disorder, psychotic disorder, being criticized, desire for attention, suicide intention, and warning signals. These variables are easily assessed and measured in routine practice. The scoring scheme is simple to calculate and interpret using user-friendly software for installation on a mobile application.

## Supporting information

**S1 Data.**
(XLS)

## Acknowledgments

The authors express their gratitude to Dr. Nattakorn Jumpathong, Director of the Khon Kaen Rachanagarindra Psychiatric Hospital, for granting permission to utilize the 506SDV dataset. Additionally, Ms. Orapin Yodkhang is acknowledged for supplying the 506SDV datasets. In addition, we express our gratitude to Dr. Kyaw Ko Jokhang Htet, Ms. Nannapat Pruphetkaew, and Ms. Jirawan Jayuphan for their expertise in statistical consultation regarding data analysis using the R software. We also acknowledge Dr. Polathep Vichitkunakorn for his valuable guidance in the construction of the web-based risk calculation model. We express our gratitude to Mr. David Patterson for his assistance in editing the English language.

## Author Contributions

**Conceptualization:** Suwanna Arunpongpaisal, Sawitri Assanangkornchai.

**Data curation:** Suwanna Arunpongpaisal.

**Formal analysis:** Suwanna Arunpongpaisal.

**Investigation:** Suwanna Arunpongpaisal.

**Methodology:** Suwanna Arunpongpaisal, Sawitri Assanangkornchai, Virasakdi Chongsuvivatwong.

**Project administration:** Suwanna Arunpongpaisal.

**Supervision:** Sawitri Assanangkornchai, Virasakdi Chongsuvivatwong.

**Validation:** Suwanna Arunpongpaisal, Sawitri Assanangkornchai.

**Visualization:** Suwanna Arunpongpaisal, Sawitri Assanangkornchai.

**Writing – original draft:** Suwanna Arunpongpaisal, Sawitri Assanangkornchai.

**Writing – review & editing:** Suwanna Arunpongpaisal, Sawitri Assanangkornchai, Virasakdi Chongsuvivatwong.

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
