## [Decision Letter · Decision Letter 0]

11 Sep 2023

PONE-D-23-18442Developing a risk prediction model for death at first suicide attempt—Identifying risk factors from Thailand’s national suicide surveillance system dataPLOS ONE

Dear Dr. Assanangkornchai,

Thank you for submitting your manuscript to PLOS ONE. After careful consideration, we feel that it has merit but does not fully meet PLOS ONE’s publication criteria as it currently stands. Therefore, we invite you to submit a revised version of the manuscript that addresses the points raised during the review process.

We look forward to receiving your revised manuscript.

Kind regards,

Pedro VS Magalhães, M.D., Ph.D.

Academic Editor

PLOS ONE

Reviewers' comments:

Reviewer's Responses to Questions

**Comments to the Author**

1. Is the manuscript technically sound, and do the data support the conclusions?

Reviewer #1: Partly

2. Has the statistical analysis been performed appropriately and rigorously? 

Reviewer #1: No

3. Have the authors made all data underlying the findings in their manuscript fully available?

Reviewer #1: No

4. Is the manuscript presented in an intelligible fashion and written in standard English?

Reviewer #1: Yes

5. Review Comments to the Author

Reviewer #1: The manuscript “Developing a risk prediction model for death at first suicide attempt - Identifying risk factors from Thailand’s national suicide surveillance system data” aimed to develop a risk-prediction model (using logistic regression) for predicting death at the first suicide attempt with data from the national suicide surveillance system from Thailand (n = 3,324 first suicide attempters). In addition, the study provided information and characteristics about first suicide attempts, including those who died at the first attempt, also pointing factors associated with suicide death in this context. The study is quite relevant and brings important data from suicide deaths and attempts from Thailand. Nonetheless, it is essential to consider specific limitations that should be worked on before considering the study for publication. Specific comments are made below:

Authors should substantially revise the text concerning English writing.

Several sentences in the introduction which seem to refer to previous work should cite the references from which they were taken from (for instance lines 46 to 56, 74-80).

The authors could expand the introduction on the description of predictive models, including a discussion on machine-learning based predictive models. Over the past years several machine learning models have been published on this topic.

The information described in lines 74-95 would be more suitable for the methods section.

In general, the introduction is very poor. Authors should include a more substantial introduction on protective and risk factors associated with suicidality, explanatory models of suicidality (for instance the one proposed by Turecki and Brent (https://pubmed.ncbi.nlm.nih.gov/26385066/), factors associated with suicide deaths in the context of suicide attempts, suicidality in Thailand, prediction of suicide-related outcomes (including traditional regression methods, and machine-learning based methods), and why is it important to study first-suicide attempts.

I miss the description of the gap in the literature that justifies the publication of this study.

Where does the equation for the calculation of individual risk of death was taken from? Was it based on prior work? It is not clear to me.

In the methods section it would be essential to describe if the dataset was split into test and training datasets. It is advisable to test the predictive model in unseen datasets, even when using traditional statistical methods, otherwise the model would be prone to overfitting, and the performance metrics could be substantially biased.

It would also be essential to report other performance metrics, mainly sensitivity and specificity. Accuracy also would be advisable.

In the example given in the end of the results section, the authors could describe whether this specific patient died as a consequence of the suicide attempt.

It is important to avoid using the term “successful suicide” for describing suicide deaths.

“Consistent with other suicidal behavior studies [30-42], we found that five interpersonal problems, namely being strongly criticized, lovesickness, quarreling with a significant person, conflicts with work colleagues, and a desire for attention, decreased the risk of successful suicide”. It is not clear to me if these variables are actually “Protective” factors against suicide deaths; for instance, lovesickness and being strongly criticized seem to be an important factor associated with suicidality. It would be important to discuss these factors further in relation to what exists in the literature. In addition, in this specific paragraph the authors should make reference to previous research.

The discussion on the possibility of using this model for the prevention of suicide deaths should be expanded. In addition, the discussion would also benefit from the discussion of existing prediction models, including machine learning models.

6. PLOS authors have the option to publish the peer review history of their article (what does this mean?). If published, this will include your full peer review and any attached files.

Reviewer #1: No

---

## [Author Response · Author response to Decision Letter 0]

22 Dec 2023

Responses to reviewers’ comments

PONE-D-23-18442

Developing a risk prediction model for death at first suicide attempt—Identifying risk factors from Thailand’s national suicide surveillance system data

PLOS ONE

Reviewer #1: The manuscript “Developing a risk prediction model for death at first suicide attempt - Identifying risk factors from Thailand’s national suicide surveillance system data” aimed to develop a risk-prediction model (using logistic regression) for predicting death at the first suicide attempt with data from the national suicide surveillance system from Thailand (n = 3,324 first suicide attempters). In addition, the study provided information and characteristics about first suicide attempts, including those who died at the first attempt, also pointing factors associated with suicide death in this context. The study is quite relevant and brings important data from suicide deaths and attempts from Thailand. Nonetheless, it is essential to consider specific limitations that should be worked on before considering the study for publication. 

Response: Thank you for these comments and for allowing us to further improve the paper.

Specific comments are made below:

1. Authors should substantially revise the text concerning English writing.

Several sentences in the introduction which seem to refer to previous work should cite the references from which they were taken (for instance lines 46 to 56, 74-80).

Response: Thank you. We have revised the text and had an English consultant who is a native English-speaking academic edit this manuscript again. We have also cited all references from which the provided information was taken. 

2. The authors could expand the introduction on the description of predictive models, including a discussion on machine-learning-based predictive models. Over the past years, several machine-learning models have been published on this topic. 

Response: Thank you. We have revised the Introduction section to include some discussion of the predictive models and the machine-learning-based models accordingly:

Lines 89-108: “A prediction model is a mathematical process used to predict future events or outcomes by analyzing patterns in a given set of risk factors [21]. A significant percentage of suicide victims interact with the medical system in the year before their death [22, 23], thus several suicide prediction systems have been developed as a result of this fact to help identify patients who should receive preventive measures. The main aim of a suicide prediction model is to enhance our capacity to predict and therefore prevent suicide, beyond the existing clinical level, by supporting clinicians in their decision-making. However, low sensitivities and low positive predictive values have led critics to argue that these tools have little clinical value and, in some cases, might do more harm than good [24, 25]. A systematic review of unique prediction models across five countries, the United States of America, Australia, the United Kingdom, Iran, and South Korea, found that the global classification accuracy of most suicide prediction models was good (≥0.80); however, their accuracy in predicting future events was extremely low (≤0.01 in most models), limiting their usability in clinical settings [26].

Some earlier studies have started with a predictor set made up of sociodemographic and clinical data taken from medical records, occasionally supplemented with a set of clinician rating scales and patient self-reports. Multivariate analyses using these data and various types of machine-learning methods have been used to develop prediction tools in more recent studies [27, 28]. However, because of the low base rate of suicides, it is hard to create a sufficiently large data set for analysis to test complex models or for the estimation of a trustworthy machine-learning algorithm.”

3. The information described in lines 74-95 would be more suitable for the methods section.

Response: Revised accordingly. However, we still explained the national suicide surveillance system in the introduction section in the sub-heading of Suicide surveillance system in Thailand at lines 124-131: 

“The Department of Mental Health of the Ministry of Public Health mandated that the Khon Kaen Rajanagarindra Psychiatric Hospital develop a National Suicide Prevention Center and suicide surveillance system in 2001. Under this system, a self-directed violence-506S (SDV-506S) case registration system with standard operating procedures was established to register all suicide attempt cases in 33 pilot community hospitals across four regions of the country, which was then expanded to all 76 provinces in 2005. The SDV 506S registration form was revised, and an online registration system was developed in 2017.” 

And we revised the method section to add the other (previous information described in lines 81-95) to Lines 147-161: 

“Each record in the SDV-506S database comprises sociodemographic and clinical characteristics of the decedents, information about suicidal behaviors, and precipitating factors. This information is obtained through medical interviews and physical and mental health examinations of the patients and their relative(s) when the patient visits the emergency department or is admitted to the hospital. The classifications of both suicide and intentional self-harm follow the International Classification of Disease, Version 10 (ICD-10) codes X60-X84, Z91.5, and R45.8, to ensure the uniformity and consistency of recorded information internationally. 

All cases of suicide attempts in public healthcare settings and suicide decedents who die at the hospital are recorded by the hospital staff after immediate care. For those who die unnaturally at home, police authorities are legally empowered to request that government medical institutions in the area perform medical forensic autopsies of suspected unnatural death cases. In cases of suspected suicide, a verbal autopsy is conducted by health professionals in the area, and their findings are recorded in the system. The revised national suicide surveillance system provides the best possible data source for studying suicide risk factors because of its large size and national representation.”

4. In general, the introduction is very poor. Authors should include a more substantial introduction on protective and risk factors associated with suicidality, explanatory models of suicidality (for instance the one proposed by Turecki and Brent (https://pubmed.ncbi.nlm.nih.gov/26385066/), factors associated with suicide deaths in the context of suicide attempts, suicidality in Thailand, prediction of suicide-related outcomes (including traditional regression methods, and machine-learning based methods), and why is it important to study first-suicide attempts.

Response: Revised accordingly.

We have included some information about the risks and protective factors for suicide, and death at first suicide attempt, prediction models and the significance of first suicide attempts.

Lines 45-87: “The causation of suicide is related to genetics, demographics, culture, geography, and personal experience [6]. Various factors can increase an individual's risk of suicide, such as being older, male, unemployed [7], having a previous suicide attempt [8], multiple childhood adversities and/or trauma, such as parental neglect, physical and/or sexual abuse [9, 10], financial difficulties [11], exposure to family violence [12], or intimate partner relationship problems [13], having a physical illness resulting in severe disability and/or chronic pain, such as traumatic brain injury victims [14], mental disorders, particularly mood disorders [15], schizophrenia [16], and alcohol/substance use disorders [17, 18]. Some prior research studies have identified several factors that are linked to a lower risk of suicide for individuals. These factors include resilience to stress, optimism, extroversion, religiosity, a strong sense of purpose and self-worth, responsibility for young children, and efficient palliative care for the elderly [6]. 

A recent umbrella review identified a wide range of individual-level risk factors for suicide mortality in the general population, most of which were from high-income countries. A previous suicide attempt and suicidal ideation and psychiatric disorders were found to be associated with a greatly increased risk of suicide mortality. Physical illnesses, such as cancer and epilepsy and sociodemographic factors (including unemployment and low education), contact with the criminal justice system, state care in childhood, access to firearms, and parental death by suicide also increased the risk of suicide mortality [3]. 

In addition, numerous risk factors for suicide might be related to one another [6], and the consequences of these factors may vary depending on the environment in which they occur. For example, losing one's work or experiencing financial difficulties can have an impact on an individual and subsequently cause their relationships to worsen. On a systemic level, these consequences can also be connected to an economic downturn. Therefore, in both research and clinical practice, identifying combinations of factors that can act in concert to increase the risk of suicide may prove to be more beneficial than listing individual ones. 

Despite abundant research on risk factors for suicide, there is a dearth of research on the factors that can predict the initial occurrence of suicidal thoughts, plans, attempts, and deaths as well as the reliability of these predictors, especially in low- and middle-income countries. A few studies on first-onset suicidal thoughts and behaviors have been conducted among college students as this group appears to be at the greatest risk of developing first-onset suicidal thoughts and behaviors. Two studies among college students found that adverse experiences such as dating violence and physical abuse before the age of 17, betrayal by someone other than the partner [19], ongoing arguments or breakup with a romantic partner, emotional abuse, parental death and lack of a reliable support system [20] were strong predictors for first-onset suicidal thoughts and behaviors. 

Moreover, a secondary data analysis among individuals aged 18-89 years old from the National Violent Death Reporting System spanning from 2005 to 2013 in the USA found that 79% of these individuals died by suicide on their first suicide attempt. Compared to those with a history of suicide attempts, those who died on their first attempt were more likely to be of the male gender, married, of African-American ethnicity, and above the age of 64. Those first-attempt decedents were also more inclined to employ extremely lethal means, less likely to have a documented mental health condition or to have communicated their intentions to others, and more likely to die if there is a physical health issue or a criminal or legal issue [5].

5. I miss the description of the gap in the literature that justifies the publication of this study. 

Response: We have added discussion on the gap in the literature at several points. For example, in Lines 41-43, we noted that more than half of suicide victims die on their first attempt but previous research focused on those with a previous suicide attempt and the risk factors for future suicidal behaviors. This implies that a sizable fraction of the population that commits suicide goes unrecognized in studies.

On Lines 70-79, we noted that there is a dearth of research on the factors that can predict the initial occurrence of suicidal thoughts, plans, attempts, and deaths as well as the reliability of these predictors, especially in low- and middle-income countries, and mentioned some available reviews on the risk factors for suicidal thoughts and behaviors among college students and the characteristics of first suicide attempters.

On Lines 112-113, we noted that little research has been conducted on the prediction tools for a first suicide attempt and its outcome.

6. Where does the equation for the calculation of individual risk of death was taken from? Was it based on prior work? It is not clear to me.

Response: We developed the risk equation by ourselves based on conventional knowledge of logistic regression as indicated in the methods section in Lines 202-208: 

“Finally, a risk prediction model relating the risk of an individual experiencing an event (death) to a set of predictors based on the first dataset was calculated using multivariable logistic regression as follows:

 Individual risk of death = exponential × (individual risk score) ÷ (1+exponential × (individual risk score), where an individual risk score = (intercept + b1×1 + b2×1 + …bpxp). b1, b2, and bp indicate regression coefficients that describe how an individual’s values of predictor variables affect the risk of death [40].

7. In the methods section, it would be essential to describe if the dataset was split into test and training datasets. It is advisable to test the predictive model in unseen datasets, even when using traditional statistical methods, otherwise the model would be prone to overfitting, and the performance metrics could be substantially biased.

Response: Revised accordingly. 

Lines 184-186: “We then split the dataset into two randomly-selected subsets, the first for developing a predictive model which included 1,824 records and the second for evaluating the model’s predictive capability with the remaining 1,500 records.” 

Lines 187-201: “For the model development, initially, bivariate associations between each independent variable and the suicidal outcomes were analyzed using the Chi-square test or Fisher’s exact test, as appropriate. We then performed multivariable logistic regression modelling separately for three predictive domains: 1) demographics, 2) underlying diseases, and 3) interpersonal and financial problems and suicide attributes. Variables with p-values less than 0.05 from all domains were then included in the combined multivariable logistic regression using the first subset of 1,824 records. All significant independent variables (p < 0.05) were included in the final risk prediction model. 

Next, to perform model validation, the prediction model derived from the first dataset was run on the second dataset of 1,500 records. The probabilities derived from the risk prediction model for all individuals were compared with those estimated using a logistic model. We then used a receiver operating characteristic (ROC) curve to assess the discriminative power of the risk prediction model, and we calculated the area under the curve (AUC) with a 95% confidence interval (CI). Other model performance metrics, including accuracy, sensitivity and specificity, were also analyzed.”

8. It would also be essential to report other performance metrics, mainly sensitivity and specificity. Accuracy also would be advisable.

Response: Revised accordingly. The accuracy, sensitivity and specificity of the model have been added in the results section in Lines 245-249: 

“When this prediction model was tested in the second dataset, the area under the ROC curve based on the second dataset (1,500 records) for the final nine-variable prediction model was 0.902 (95% CI: 0.886, 0.917), and the optimum cut-off point was 52.16 with a sensitivity of 84.65%, a specificity of 82.66%, and an accuracy of 83.63%, indicating a high discriminative ability. (Fig 1 and Table 4)

9. In the example given in the end of the results section, the authors could describe whether this specific patient died as a consequence of the suicide attempt. 

Response: The illustration given at the end of the results section was only a hypothetical patient to show how the equation could be used to calculate the risk score and probability of death on the first attempt. It was not a real patient.

10. It is important to avoid using the term “successful suicide” for describing suicide deaths.

Response: Thank you for your suggestion. We have changed the term “successful suicide” to another term as appropriate. We used the term “ to prevent subsequent suicide” instead of “ to prevent successful future” in Lines 345-346”: “Moreover, it is more convenient and uncomplicated for an emergency physician to evaluate the first suicide attempt and choose appropriate measures to prevent subsequent suicide.” 

11. “Consistent with other suicidal behavior studies [30-42], we found that five interpersonal problems, namely being strongly criticized, lovesickness, quarreling with a significant person, conflicts with work colleagues, and a desire for attention, decreased the risk of successful suicide”. It is not clear to me if these variables are actually “Protective” factors against suicide deaths; for instance, lovesickness and being strongly criticized seem to be an important factor associated with suicidality. It would be important to discuss these factors further in relation to what exists in the literature. In addition, in this specific paragraph the authors should make reference to previous research.

Response: Revised accordingly. The discussion of these findings has been revised according to the new results.

Lines 317- 334: “Our findings indicate that two specific interpersonal difficulties, namely experiencing severe criticism and desire for plentiful attention, were negatively associated with the fatality outcome among the first suicide attempters. Previous studies found relationships between some interpersonal factors, for example, perceived criticism [51], high level of expressed emotion [52], bullying [53], and cyber-victimization [54] with suicide ideation and attempts. These links are often explained by the interpersonal theory of suicide which emphasizes the mediation role of thwarted belongingness and perceived burdensomeness [55, 56]. These relationships are also in line with our study, in which we also found that the experience of being strongly criticized, blamed, or intimidated was common among our subjects (32.67% of all first suicide attempters) and more frequent in the surviving attempters (57.7%) than in the decedents (42.3%). Similarly, a desire for plentiful attention was higher among the surviving attempters (67.5%) than the decedents (32.5%), making it a risk factor for non-fatal suicidal behaviors. Suicide attempts may be an expression of feelings, like emotional pain, sadness, abandonment, or anger, in response to criticism or a desire for plentiful attention from a significant person; thus, their role in the risk prediction model can be a negative risk factor for death. If the significant persons recognize and respond appropriately, suicide attempters may change their suicidal thoughts and want to continue living. Furthermore, cognitive reappraisal or the development of healthy coping skills to deal with criticism and a desire for plentiful attention may be especially important for these people.” 

12. The discussion on the possibility of using this model for the prevention of suicide deaths should be expanded. In addition, the discussion would also benefit from the discussion of existing prediction models, including machine learning models.

Response: Thank you for your comments. We have added some discussion on this point in Lines 317-334: “Our suicide risk-prediction model may have an advantage over traditional tools in Thailand, like the 8 Questions (8Q) for the Suicidality Test [57]. Moreover, it is more convenient and uncomplicated for an emergency physician to evaluate the first suicide attempt and choose appropriate measures to prevent subsequent suicide. However, the model requires external validation to calibrate and confirm its discriminative ability.” 

Line 360-381: “With the high availability of electronic health records (EHR) in modern medical practice, studies of machine-learning methods have grown rapidly as suicide prediction methods [58]. Although the use of machine learning in suicide research is promising for improving the prediction and prevention of suicidal thoughts and behaviors, these machine learning methods require voluminous data to first train the model and a different dataset to evaluate the model. Our national suicide surveillance system has only developed an electronic online system in 2017 and records of the first suicide attempters are still small. Besides, our aim for this research was not only to develop a prediction model but also to understand and interpret the relationship between predictors and the outcome variable. Therefore, we used logistic regression to develop the prediction algorithm. In the future when our surveillance system is well established and contains a large amount of data, a machine-learning prediction model can be applied for further improving the prediction tool. 

This predictive model for mortality in individuals making their first suicide attempt can be extensively utilized in emergency departments and outpatient clinics, including counselling services and inpatient facilities. To enhance the efficacy of suicide prevention initiatives, it is advisable to facilitate greater engagement and intervention for those who have made their first suicide attempt, particularly those with a higher likelihood of actually committing suicide, using the mobile application of this prediction model. To ensure the accuracy and reliability of the predictor model for mortality in individuals making their first suicide attempt, further investigation is required utilizing larger datasets, advanced machine learning techniques, and rigorous validation methodologies. Moreover, approaches tailored for individual subjects should be implemented in future studies.”

---

## [Decision Letter · Decision Letter 1]

16 Jan 2024

Developing a risk prediction model for death at first suicide attempt—Identifying risk factors from Thailand’s national suicide surveillance system data

PONE-D-23-18442R1

Dear Dr. Assanangkornchai,

We’re pleased to inform you that your manuscript has been judged scientifically suitable for publication and will be formally accepted for publication once it meets all outstanding technical requirements.

Kind regards,

Pedro VS Magalhães, M.D., Ph.D.

Academic Editor

PLOS ONE

Additional Editor Comments (optional):

Reviewers' comments:

Reviewer's Responses to Questions

**Comments to the Author**

1. If the authors have adequately addressed your comments raised in a previous round of review and you feel that this manuscript is now acceptable for publication, you may indicate that here to bypass the “Comments to the Author” section, enter your conflict of interest statement in the “Confidential to Editor” section, and submit your "Accept" recommendation.

Reviewer #1: All comments have been addressed

2. Is the manuscript technically sound, and do the data support the conclusions?

Reviewer #1: Yes

3. Has the statistical analysis been performed appropriately and rigorously? 

Reviewer #1: Yes

4. Have the authors made all data underlying the findings in their manuscript fully available?

Reviewer #1: Yes

5. Is the manuscript presented in an intelligible fashion and written in standard English?

Reviewer #1: Yes

6. Review Comments to the Author

Reviewer #1: After reading the response to my previous comments concerning the first round of peer review, it seems that the authors adressed all my suggestions.

7. PLOS authors have the option to publish the peer review history of their article (what does this mean?). If published, this will include your full peer review and any attached files.

Reviewer #1: No

---

## [Editor Report · Acceptance letter]

30 Jan 2024

PONE-D-23-18442R1 

PLOS ONE

Dear Dr. Assanangkornchai, 

I'm pleased to inform you that your manuscript has been deemed suitable for publication in PLOS ONE. Congratulations! Your manuscript is now being handed over to our production team.

Kind regards, 

on behalf of

Professor Pedro VS Magalhães 

Academic Editor

PLOS ONE